# Multi-Sensor Platform for Predictive Air Quality Monitoring

**DOI:** 10.3390/s23115139

**Published:** 2023-05-28

**Authors:** Gabriele Rescio, Andrea Manni, Andrea Caroppo, Anna Maria Carluccio, Pietro Siciliano, Alessandro Leone

**Affiliations:** National Research Council of Italy, Institute for Microelectronics and Microsystems, 73100 Lecce, Italy; andrea.caroppo@cnr.it (A.C.); annamaria.carluccio@imm.cnr.it (A.M.C.); pietroaleardo.siciliano@cnr.it (P.S.); alessandro.leone@cnr.it (A.L.)

**Keywords:** indoor air quality, carbon dioxide, HVAC system, deep learning, forecasting

## Abstract

Air quality monitoring is a very important aspect of providing safe indoor conditions, and carbon dioxide (CO${}_{2}$) is one of the pollutants that most affects people’s health. An automatic system able to accurately forecast CO${}_{2}$ concentration can prevent a sudden rise in CO${}_{2}$ levels through appropriate control of heating, ventilation and air-conditioning (HVAC) systems, avoiding energy waste and ensuring people’s comfort. There are several works in the literature dedicated to air quality assessment and control of HVAC systems; the performance maximisation of such systems is typically achieved using a significant amount of data collected over a long period of time (even months) to train the algorithm. This can be costly and may not respond to a real scenario where the habits of the house occupants or the environment conditions may change over time. To address this problem, an adaptive hardware–software platform was developed, following the IoT paradigm, with a high level of accuracy in forecasting CO${}_{2}$ trends by analysing only a limited window of recent data. The system was tested considering a real case study in a residential room used for smart working and physical exercise; the parameters analysed were the occupants’ physical activity, temperature, humidity and CO${}_{2}$ in the room. Three deep-learning algorithms were evaluated, and the best result was obtained with the Long Short-Term Memory network, which features a Root Mean Square Error of about 10 ppm with a training period of 10 days.

## 1. Introduction

Indoor air quality (IAQ) monitoring has been continuously investigated, as higher levels of pollutants have often been found indoors than outdoors [1,2]. Various parameters present in environments, such as particulate matter (PM) and carbon dioxide (CO${}_{2}$), have been analyzed. In particular, indoor CO${}_{2}$ accumulation as a contributing factor to bad indoor air has been the focus of many discussions on building ventilation and IAQ [3]. Prolonged exposure to high CO${}_{2}$ concentrations in indoor spaces has harmful effects on human health; in fact, it can cause headache, fatigue, drowsiness, nausea, memory loss, sleep-cycle disorders, etc. [4,5]. For CO${}_{2}$ concentrations exceeding 700 ppm, symptoms of sick building syndrome (SBS) have been observed; while, for values exceeding 1000 ppm, inhibition of cognitive performance in school children has been reported. Further dysfunction was recorded for ppm values between 1000 and 5000. This led to the identification of thresholds of indoor CO${}_{2}$ concentrations reflecting good (CO${}_{2}$ < 1000 ppm), moderate (CO${}_{2}$ between 1000 and 1500 ppm) and poor (CO${}_{2}$ > 1500 ppm) IAQ. Indoor CO${}_{2}$ concentrations and associated health risks have become increasingly important due to the growing amount of time spent indoors since COVID-19, with the spread of teleworking and smart working. In fact, these work modalities are leading more and more people to use their homes as their working environment; therefore, monitoring of air quality parameters has become essential, in order to be able to detect suboptimal conditions for the individual’s health and to be able to appropriately activate ventilation systems. In these contexts, in addition to the sources of CO${}_{2}$ production inherent in the home environment (cooking, heating systems, etc.), the rate of CO${}_{2}$ generation by building occupants becomes relevant. It varies with the individual characteristics of occupants (gender, age, weight, body composition, fitness level, etc.), which influence energy expenditure and the ratio of O${}_{2}$ consumed to CO${}_{2}$ produced [6]. In addition to CO${}_{2}$, bioeffluents from human occupants are also the cause of indoor air problems; however, in this paper, we will focus only on the analysis of CO${}_{2}$ widely used to characterize IAQ conditions in buildings and the adequacy of outdoor air ventilation.

Another effect of the pandemic has been to encourage the performing of physical activity at home, especially among adults [7]. This simultaneously leads to an increase in the amount of CO${}_{2}$ produced indoors and the need for a high amount of Oxygen for those who exercise. Therefore, it is important for CO${}_{2}$ monitoring systems to be able to recognize the trend of increasing pollutants early enough to activate ventilation systems and maintain the condition of optimal air quality without having heat loss. Manual ventilation control often fails to eliminate concentration peaks; on the other hand, an automatic system capable of accurately predicting CO${}_{2}$ levels can prevent its sudden rise and appropriately drive the ventilation system, avoiding wasted energy and ensuring people’s comfort.

In recent years, several studies on indoor CO${}_{2}$, which have achieved high levels of CO${}_{2}$ prediction accuracy through various artificial-intelligence techniques, have been published. They analyse various environmental parameters such as temperature, CO${}_{2}$ and humidity through statistical models and machine-learning approaches, including Artificial Neural Networks (ANNs), Linear Regression (LR) models and Decision-Tree (DT)-based models [8,9]. Such works do not perform a forecast of CO${}_{2}$ concentration in a future time window, and few works are found in the literature with this purpose [10,11,12,13,14,15]. In these works, performance maximization is typically achieved using a significant amount of data collected over many days or months for algorithm training. This, besides being very costly, may not respect a real-world scenario where the habits of the home occupants or the conditions of the environment may change over time. The Kallio research [14] monitored five days’ historical data to forecast the CO${}_{2}$ concentrations of two days in the following week and achieved good performance with an ANN, but the data was limited to nine hours per day with a resolution of one hour. Therefore, since CO${}_{2}$ concentrations can change significantly over the course of an hour, the data set is not accurate enough to drive an adaptive ventilation system. To achieve this, it is relevant that the control system has as input for training a time window of updated data. Segala et al. [15] addressed this problem by developing a store CO${}_{2}$ prediction by a Deep-Learning (DL) system (a Convolution Neural Network) which updates over time and considers a limited amount of data for training. Such a system analyzes temperature, CO${}_{2}$ and humidity data and obtains a prediction error of around 15 ppm after using a week of data for training the model and about 10 ppm after 30 days. To the best of our knowledge, no work in the literature has dealt with developing something similar in the domestic context. In such an environment, higher performance can be achieved by adding an accurate assessment of the occupants’ physical activity level to the monitoring of environmental signals. Activity level can be assessed in several ways using environmental (e.g., 3D vision camera, Passive InfraRed sensor, etc.) or wearable (e.g., accelerometers, gyroscopes, etc.) sensors. The former type of sensors is non-obtrusive to individuals, but requires the ad-hoc design of all monitoring rooms; this can be costly and complex. In contrast, through wearable devices, a more cost-effective solution can be achieved, at the expense of a higher level of invasiveness; however, this problem has been reduced in recent years, due to the spread of increasingly miniaturized and easy-to-use sensor devices (e.g., smartwatches, smartphones, sensorized t-shirts, etc.).

In the present work, a CO${}_{2}$ forecasting system for a home environment is presented which monitors the temperature, humidity, CO${}_{2}$ and physical activity of the users by means of ubiquitous sensors, which can detect data in real time. The design and testing of the system was carried out by considering a real-world case study in a household inhabited by a family, in which members perform smart working and exercise activities in a dedicated environment. Three Deep-Learning algorithms were identified for CO${}_{2}$ concentration assessment which allow high accuracy values (about 10 ppm maximum error) with a 10-day training and without the room-specific details, such as dimensions and volume. Preliminary tests were carried out considering only one room in the house occupied by one user at a time. The system can be replicated in several rooms and take into account the presence of all family members in the room. A methodological and performance comparison with other recent publications is given in Table 1.

The paper is structured as follows. Section 2 reports an overview of the hardware architecture and algorithmic framework for CO${}_{2}$ forecasting. Performance results of the algorithms are described in Section 3. Finally, Section 4 shows both our conclusions and discussions of some ideas for future work.

## 2. Materials and Methods

### 2.1. Hardware Architecture

The architecture of the proposed system consists of the following parts:Wireless air quality sensor capable of acquiring the following environmental parameters: particulate matter (PM 1, PM 2.5, PM 10), CO${}_{2}$, VOC, atmospheric pressure, temperature, humidity;Sensorized T-shirt to monitor torso movements, breathing and heart rate;Embedded Personal Computer (PC) for data collection and processing.

The air quality sensor is the UPAI3-CPVTHA model manufactured by Upsens [16] and shown in Figure 1. It is equipped with 4 LEDs signaling the quality of 4 selected parameters and uses the MQTT protocol and WiFi connection to send data. The CO${}_{2}$ sensor has a resolution of 1 ppm and an accuracy of ±5% [17]. It also integrates pressure, humidity and temperature (in the range −10 ${}^{{}^{\circ}}$C to 50 ${}^{{}^{\circ}}$C) compensation to keep the measurement within the stated accuracy. The time interval of data acquisition and transmission can be set according to design requirements.

The Wearable Wellness System (WWS) T-shirt, manufactured by Smartex [18], integrates a piezoresistive sensor for breathing-rate assessment and two textile electrodes for heart-rate measurement. These sensors are connected to an electronic board (dimensions 51 mm × 62 mm × 14 mm, weighting about 50 g) placed in a properly made housing in the garment, at the level of the torso, as shown in Figure 2. In addition to addressing data acquisition and transmission, the board integrates a triaxial, DC-coupled accelerometer sensor suitable for the analysis of torso movements. The garment uses Bluetooth wireless protocol in a range of up to 10 m, and multisensory information is sampled at 25 Hz, a frequency adequate to assess an individual’s postures and activity level. The battery life, in streaming mode, is approximately 8 h.

Although the two sensor systems acquire numerous parameters, in order to reduce the computational cost, it was decided to analyse the best performing signals, described in the literature, for the analysis of CO${}_{2}$ trends over time. Specifically with regard to the air quality analysis system, CO${}_{2}$, temperature and humidity were taken into consideration. While, for the t-shirt, only the activity level was calculated through the analysis of accelerometer signals using an algorithm developed for a similar accelerometer system [19]. It calculates the motion activity level with an accuracy of about 96% through posture detection and gait velocity analysis..

The data of interest were stored and processed through the Lenovo ThinkStation i5 embedded PC with 8GB RAM, which features enough computational capacity to analyze the data in real time, including using advanced artificial-intelligence software. The PC is equipped with both Bluetooth and Wifi wireless connections, so it is able to communicate with the two systems (directly with the wearable T-shirt, via cloud, with the air quality sensor).

The scheme of the hardware architecture is shown in Figure 3.

### 2.2. Data Acquisition

The experimental study was conducted in a room of approximately 18 square metres by a height of 3 m, equipped with a hot/cold air conditioner, located inside a residential house, in a small village (Borgagne in the province of Lecce, Italy). The acquisition was carried out for about 15 days (from 2 March 2023 to 17 March 2023). The room has a workstation equipped with a PC for smart working and a treadmill and weight bench for performing physical activities. The room is equipped with (a) a window facing outdoors, made of aluminium with a high degree of thermal insulation, measuring 280 cm × 150 cm, and (b) an interior wooden door, not perfectly insulated, measuring 250 cm × 120 cm. Inside the room, at a height of about 1.5 m from the ground, the air quality analysis sensor described in paragraph “Hardware Architecture” was installed. The sensor was properly positioned away from windows or air vents that could affect the measurement of the parameters of interest.

For data acquisition, 3 people (2 women and 1 man) in good physical health, and aged between 31 and 42 were involved. Each participant in the acquisition campaign performed normal daily activities by occupying the room on alternate days so that there was only one occupant in the room at a time.

Data was recorded for 15 consecutive days for 24 h as follows:Ten days for an average of about 8 h per day for smart working;Twelve days for an average of about 2 h per day for physical activity;The remainder of the recording covered the room under non-inhabited conditions.

The door and window were almost always closed. Only the window was opened for a duration of about 10 min when the CO${}_{2}$ levels in the room were unsafe for the occupant’s health.

Participants were asked to read and sign the consent and wear the WWS smart T-shirt to record their activity level. The air quality sensor was active 24 h a day for all days, and the cadence of data acquisition and sending to the cloud platform was set to 1 min in order to have a more detailed analysis of the temporal trend of the parameters of interest. Meanwhile, the accelerometer data, coming from the T-shirt, was sent directly to the PC described in the “Hardware architecture” section in streaming mode. Subsequently, the activity level was calculated every 5 s using a software implemented in the Python programming language for a similiar accelerometer device [20]. This software is also involved in (a) reading the data from the environmental sensors, and (b) aligning the time and processing all parameters of interest appropriately.

### 2.3. Methodology

Since the latest algorithmic frameworks, based on deep-learning architectures, can be deployed on low-cost commercial hardware architectures (even those not integrating GPUs), in our proposed approach three DL architectures were considered for performance comparison on the forecasting model: (1) a one-dimensional Convolutional Neural Network (1D-CNN) [21], (2) Long Short-Term Memory (LSTM) [22] and (3) a Recurrent Neural Network (RNN) [23]. Each of the 1D-CNN, LSTM and RNN models often show good performance on multivariate time-series data [24,25,26,27].

As a type of CNN, the 1D-CNN has the basic characteristics of CNN and is suitable for the time-series processing of sensor data. By using convolution processes, it can automatically evaluate and extract features from a single spatial dimension, thereby discovering intricate patterns in the data. A schematic representation of this type of architecture, customized with respect to the type of input/output data considered in this work, is depicted in Figure 4. A significant advantage of 1D-CNN over widely used neural networks is that it learns features of a signal by considering local information instead of the whole signal in each network layer, decreasing computational load required and thus favouring their use even on processing devices with limited power (i.e., embedded platforms) [28]. To process one-dimensional data, 1D-CNN employs one-dimensional convolution layers, pooling layers, dropout layers and activation functions. In addition, to configure the network, the following hyper-parameters are used: the filter size, the subsampling factor of each layer and the number of neurons in each layer. Information on the tuning of previously introduced parameters will be detailed in the following section.

On the other hand, RNNs (Figure 5) are the best algorithms for dealing with time series [29], because they analyse the input data sequence iteratively and explicitly describe the sequence of the input data. In fact, RNNs loop over the input data sequence, keeping an internal model of the information they are processing, built from past information and constantly updated as new information arrives, unlike other neural network architectures that process the input data at once (i.e., CNN). Another added value is that RNNs often need fewer layers to complete a task than other neural network architectures because of their recurring nature. Although, from a theoretical point of view, RNN is able to handle such long-term dependency problems, the weighting matrix will continue to multiply repeatedly with the previous output, in accordance with the length of the time interval (and, in this case, the data to be referred to will be at a greater temporal distance). The vanishing gradient and exploding gradient problem [30] will result from this. Consequently, it is preferable to use, in some contexts, LSTM networks to address this issue and make improvements.

LSTM is an evolution of the RNN architecture which started to be used in the mid-2000s in applications such as stock-market forecasting [31] and speech recognition [32]. More recently, LSTM was used for COVID-19 pandemic new-cases prediction [33,34]. LSTM network is composed of a series of cells, each of which consists of a cell state and input, forget and output gates which make use of several activation functions (generally, sigmoid and tanh). Specifically, with the “forget gate”, the architecture decides which information should be kept and which should be discarded, whereas in the “input gate”, it updates the cell state. In the “output gate”, LSTM decides what the next hidden state should be and, finally, with “cell state”, it acts as a highway that transports relative information along the sequence chain. The main distinction between an LSTM cell and a conventional RNN cell is that an LSTM cell has a memory unit which has long-term memory capabilities. A forget gate, which can selectively forget information that is no longer relevant, is used to do this. In Figure 6, the structure of LSTM is reported.

### 2.4. Neural-Network Architectures

Following the same approach as [15], the model was updated over time using a sliding window to avoid the inclusion of old data which could degrade the accuracy of the forecasts. This work, however, unlike [15], also considers activity level as an input variable for networks to improve prediction accuracy.

The considered neural network architectures were designed using TensorFlow (version 2.11.0), and Python (version 3.7.9). 1D-CNN architecture consists of the following layers:Input layer: each sample includes the values of the input variables (CO${}_{2}$, temperature, humidity and activity level) for each minute of acquisition. The input values are normalised before loading the neural network. In particular, the input values are scaled so that they lie in the range given on the training set, in our case between zero and one [35]. The scaling is given by the following equation:
$$X_{scaled}=\frac{\left(X-min\right)}{\left(max-min\right)}$$
where minimum and maximum values are related to the x-value to be normalised.One-dimensional Convolutional layer: It is used for the analysis and extraction of features along the temporal axis of the inputs. To extract non-linear feature patterns from the data, the standard rectified linear activation function (i.e., ReLU) is employed.Max Pooling layer: Its purpose is to learn the most useful information from the feature vectors by subsampling the output matrix from the previous layer.Flatten layer: The input matrix is reshaped to produce a one-dimensional feature vector to generate predictions from the output layer.Output layer: The output of this fully connected linear layer is a single neuron to forecast the CO${}_{2}$ value for the next minute.

RNN architecture is composed in the aftermentioned layers:Input layer: like CNN.Three RNN layers: After the input layer, those three layers are present to improve the performance of our model and provide reasonable results compared to conventional neural-network models.Three Dropout layers: A dropout layer was added after each RNN layer in order to improve the forecast accuracy and compensate overfitting.Output layer: as for CNN.

Finally, LSTM has the following layers:Input layer: like the two previous architectures.Three LSTM layers: as described for RNN, these three layers increase performance in CO${}_{2}$ forecasting.Three Dropout layers: As with the RNN, a dropout layer was added after each LSTM layer to enhance forecast values.Output layer: like the two previous architectures.

To obtain the optimal parameters for each DL architecture, a random search technique [36] was used. This solution does not require the gradient of the problem to be optimized, but instead defines a search space of hyperparameter values and randomly samples points in that domain. Neural networks use this technique for hyperparameter optimization. Table 2 shows the ranges of the considered parameters for each architecture while Table 3 contains the selected parameters for each architecture and Figure 7 shows the structure for each architecture, in particular, (a) 1D-CNN, (b) RNN and (c) LSTM. In particular, for 1D-CNN, Conv1D output (None, 1, 128) means that it has 128 hidden layers, while dense output (None, 20) indicates that it has 20 hidden layers, and so on for the other architectures considered.

## 3. Results and Discussion

To validate the proposed approach, a series of tests was executed to verify the accuracy of the CO${}_{2}$ value forecast. Our experiments were run on an embedded PC with an Intel Core i5 processor and 8 GB RAM. The performances of the proposed architectures were estimated using the root mean square error (RMSE) as metric, defined as:(1)$$RMSE=\sqrt{\frac{1}{N}\sum_{i=1}^{N}\left(y_{i}-{\bar{y}}_{i}\right)}$$
where $y_{i}$ is the real CO${}_{2}$, ${\bar{y}}_{i}$ is the predicted CO${}_{2}$ and *N* is the number of minutes to forecast. In addition, to demonstrate the goodness of the proposed approach, the normalised RMSE (NRMSE), obtained by means of the following expression, was also adopted as a metric:(2)$$NRMSE=\frac{RMSE}{\max(y)-\min(y)}$$

Preliminary tests were conducted to assess the amount of data to be used to train the analyzed models, to guarantee the accuracy of the predictions while preserving acceptable training times. To achieve this, the configuration described in Section 2.2 was considered, in particular, a room was used, with a person inside who may or may not have been physically active.

Figure 8, Figure 9 and Figure 10 show the obtained results for these preliminary tests demonstrating the goodness of the proposed approach. In particular, in the figures, the RMSE and the training time varying with the days of acquisition with and without the Activity Levels obtained, respectively, for 1D-CNN (Figure 8a,b), RNN (Figure 9a,b) and LSTM (Figure 10a,b) are reported. Firstly, it can be seen that the accuracy of the prediction clearly improves with 7 days of acquisition for training with the RMSE dropping from around 50 ppm to around 18 ppm indicating that, increasing the number of days of observation, the analysed networks are able to learn the considered features. The best results, in terms of forecast accuracy, are obtained after 10 days of acquisition with an RMSE value of about 10 ppm, which guarantees an excellent accuracy for the regulation of HVAC systems. It can also be seen from the figures that for all the analyzed networks, and especially for 1D-CNN and LSTM, considering more than 10 days of acquisition did not produce any significant improvement in terms of RMSE, while training time tended to increase exponentially. Therefore, it can be deduced that 10 days of acquisition is a good compromise between prediction accuracy and training time, especially from the point of view of experimentation on low-cost embedded platforms with reduced computational processing capacity. It is evident from the graphs that activity levels provide a significant increase in performance in terms of RMSE (e.g., for LSTM 10.31 ppm vs. 21.18 without Activity Level) without affecting training time. Moreover, as described in [20], the computational load of the wearable system’s processing software is low, guaranteeing real-time operation, even on low-cost computing platforms.

To extract more information from the described approach, the obtained results were analysed at a more granular level. In particular, Table 4 shows RMSE and NRMSE during periods when the room was disabled versus periods of work versus periods of physical activity for the three investigated architectures.

In order to confirm the effectiveness of the methodology with the inclusion of the Activity Levels, the trend of the CO${}_{2}$ concentration forecasting for 15 min with and without Activity Level for the aforementioned three neural networks was analyzed. In particular, the 15-min forecast for 1D-CNN with and without Activity Level obtained an RMSE of 10.42 ppm and 21.34 ppm, respectively, while for RNN RMSE was 15.87 ppm and 25.45 ppm, respectively, and, finally, for LSTM, RMSE forecast was 10.31 ppm vs. 21.18 ppm, respectively. For brevity, only the trend of LSTM was plotted in Figure 11 and Figure 12. Finally, the trend of the CO${}_{2}$ concentration forecasting for a longer time (120 min) for the three analysed networks (Figure 13, Figure 14 and Figure 15) is shown, demonstrating the forecast accuracy. For the sake of brevity, the prediction plots are only presented with Activity Levels. Finally, Table 5 presents RMSEs for each neural network both with and without Activity Levels for a 120-min time prediction.

In conclusion, as can be seen from the trends in Figure 13, Figure 14 and Figure 15 and also from Table 5, LSTM has a better prediction accuracy than the 1D-CNN network, while RNN performs the worst. Analysing the training times, 1D-CNN is the one that requires less time than the other two neural networks considered, but the LSTM network’s training times are also acceptable and compatible with running on low-cost computing platforms. For this reason, the LSTM network was chosen for the present work.

In addition, a series of tests was carried out in order to verify the importance of the various inputs. The results are shown in Table 6. As can be seen, the error committed is greater when removing both Temperature and Humidity, while when removing only one input at a time, the error committed is similar whether removing only Temperature or only Humidity.

Finally, the performance of the proposed solution varying the participants in the test were considered. In particular, the periods in which the various users were engaged in physical activity were analyzed, and RMSE was calculated vs the users and considered networks. The results are shown in Figure 16 and, as can be seen, the results are similar for user 1 and 3 and slightly higher on average for user 2, who was subjected to more intense physical activity than the other users.

## 4. Conclusions

This paper presents a software–hardware framework for the evaluation of CO${}_{2}$-concentration trends in a residential environment suitable for driving an automatic HVAC control system. High CO${}_{2}$ concentrations can cause problems for the health and comfort level of occupants and this phenomenon has become more acute in recent years, in which the development of houses with high thermal insulation has brought significant improvements in terms of energy savings, but has hindered a natural exchange of air with the outside environment. This can lead to a significant and rapid growth in the amount of CO${}_{2}$ in the home and, therefore, a continuous assessment of air quality is necessary in order to provide timely and efficient intervention. To achieve this, it is important that the system acts in an adaptive way, considering a reduced historical time window, which is useful for accurate knowledge of recent environmental conditions, and is able to predict CO${}_{2}$ trends with high accuracy in anticipation. Furthermore, the use of limited datasets could allow the implementation of the proposed approach on low-cost embedded platforms, supporting a wider deployment.

The developed system uses low-cost and minimally invasive environmental and wearable sensors. Both sensors are equipped with wireless communication which enables remote, real-time processing and allows monitoring of environmental parameters (temperature, humidity, CO${}_{2}$) and physical activity. The latter parameter analysis was verified to improve prediction accuracy appreciably, at the cost of an increase in system complexity from a hardware point of view. The data were analysed by testing three DL algorithms (1D-CNN, RNN, LSTM), and the best result was obtained with the LSTM network, which features a low RMSE value (about 10 ppm) compatible with HVAC systems with a training period of 10 days, which is shorter than other works in the literature.

The system was tested in a real-world environment, however, considering a single room used for smart working and exercising. Future developments include the monitoring of more than one room occupied by multiple people at the same time. Moreover, a less invasive t-shirt system using, for example, a smartwatch with Wifi communication, will be developed and tested for activity assessment.

## Figures and Tables

**Figure 1 sensors-23-05139-f001:**
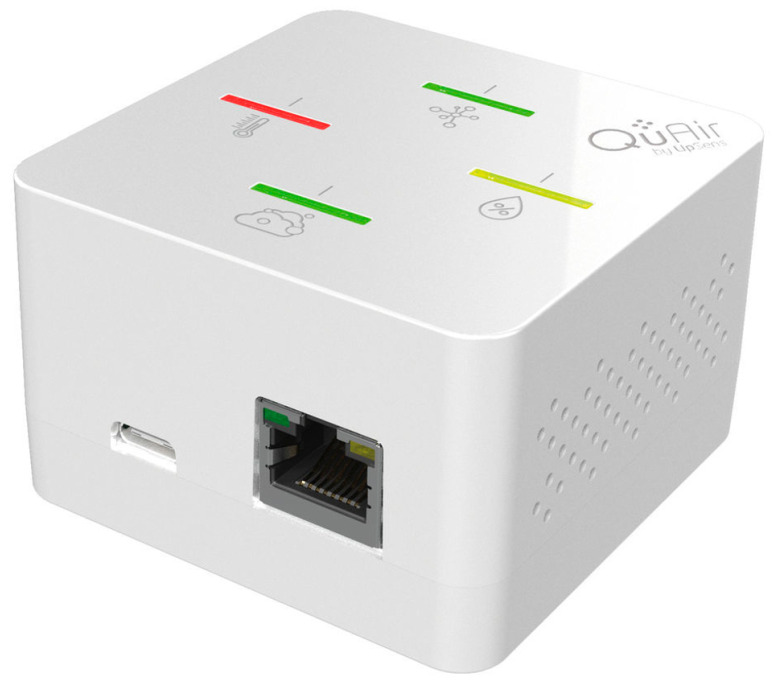
UPAI3-CPVTHA air quality sensor produced by Upsens.

**Figure 2 sensors-23-05139-f002:**
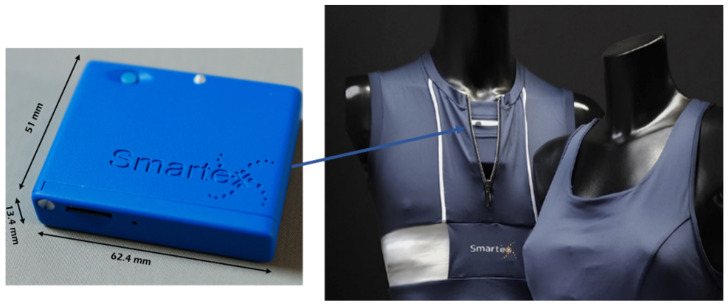
WWS sensorized t-shirt produced by Smartex.

**Figure 3 sensors-23-05139-f003:**
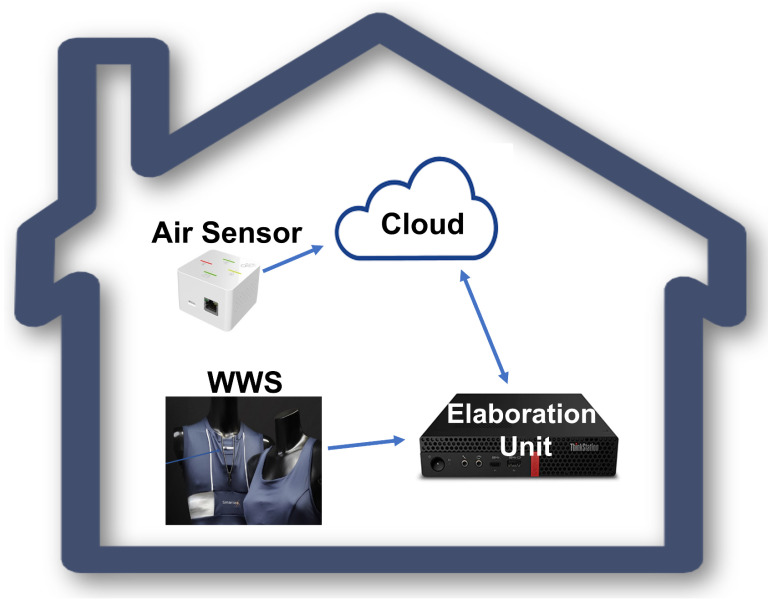
Hardware architecture overview.

**Figure 4 sensors-23-05139-f004:**
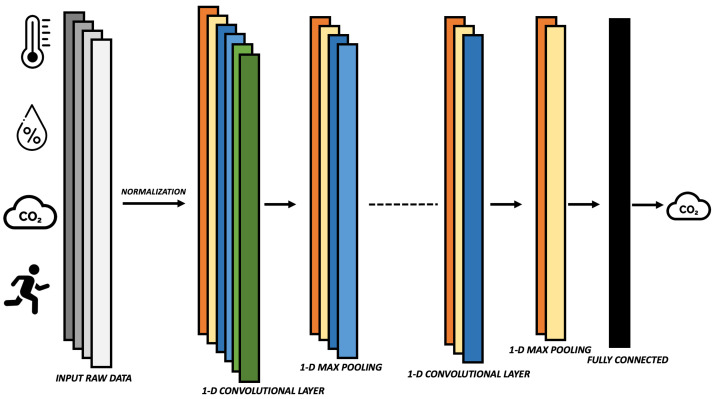
A typical representation of 1D-CNN for CO${}_{2}$ forecasting using as input temperature, humidity, CO${}_{2}$ and activity level.

**Figure 5 sensors-23-05139-f005:**
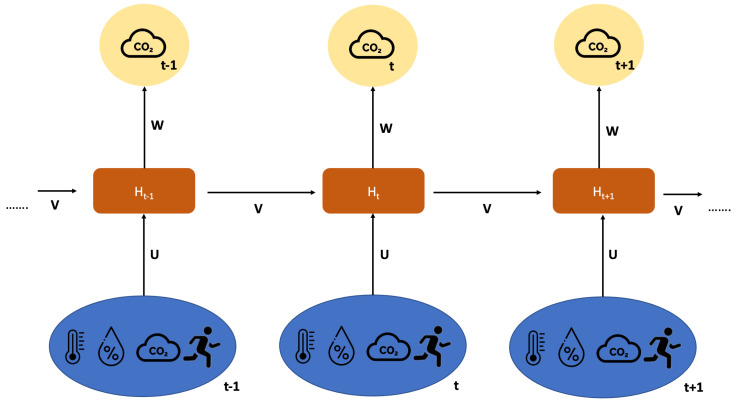
RNN representation through logic blocks. The **bottom** is the input state; **middle**, the hidden state; **top**, the output state. U, V, W are the weights of the network.

**Figure 6 sensors-23-05139-f006:**
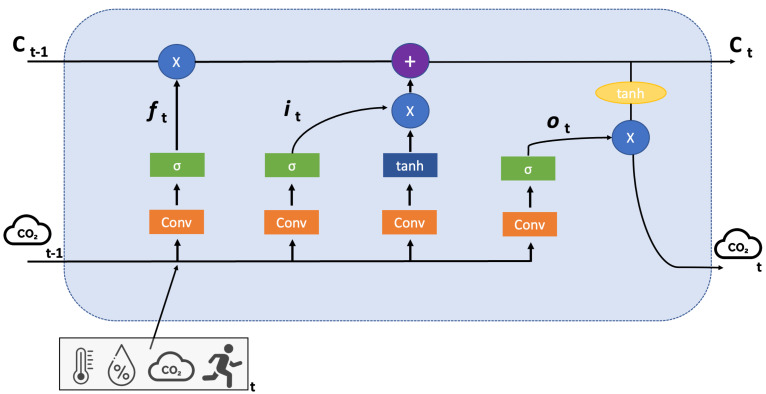
A typical representation of LSTM architecture. The new memory C${}_{t}$ and the output at time *t* (the predicted value of CO${}_{2}$) will be generated by updating the internal memory C${}_{t-1}$ according to the current input at time *t* (temperature, humidity, CO${}_{2}$ and activity level) and the previous CO${}_{2}$ output value at time t−1.

**Figure 7 sensors-23-05139-f007:**
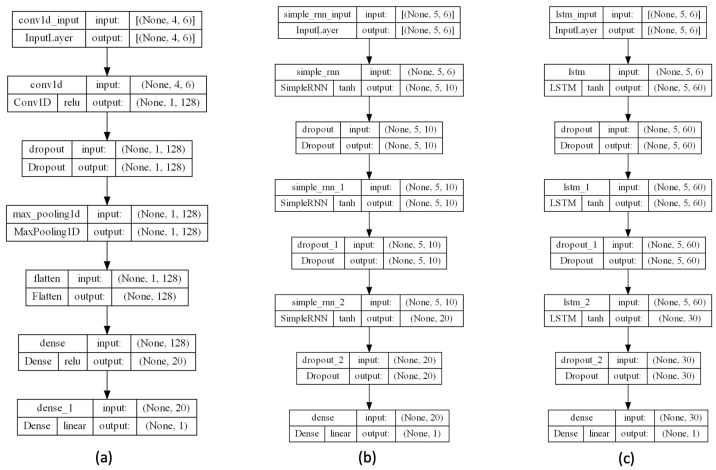
The designed architectures for considered neural networks: (**a**) 1D-CNN, (**b**) RNN, and (**c**) LSTM.

**Figure 8 sensors-23-05139-f008:**
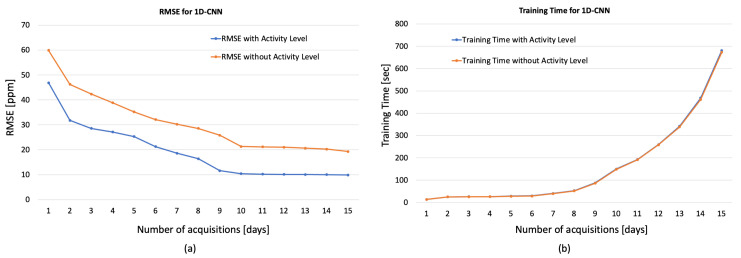
RMSE (**a**) and training time (**b**) for 1D-CNN varying the days of acquisition with and without the Activity Levels.

**Figure 9 sensors-23-05139-f009:**
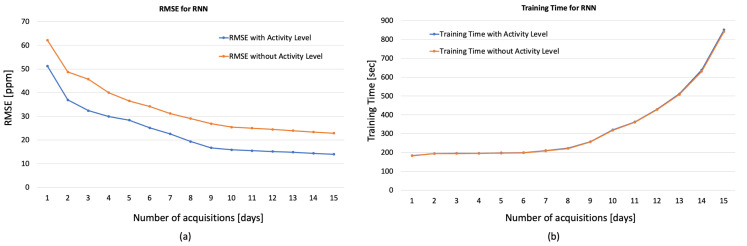
RMSE (**a**) and training time (**b**) for RNN varying the days of acquisition with and without the Activity Levels.

**Figure 10 sensors-23-05139-f010:**
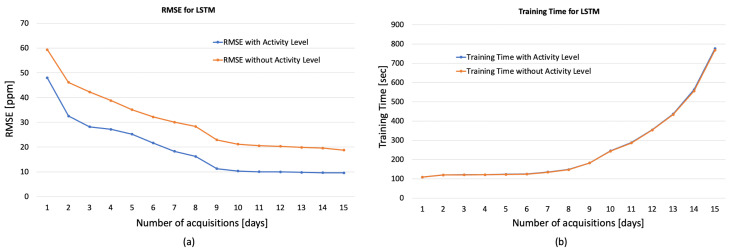
RMSE (**a**) and training time (**b**) for LSTM varying the days of acquisition with and without the Activity Levels.

**Figure 11 sensors-23-05139-f011:**
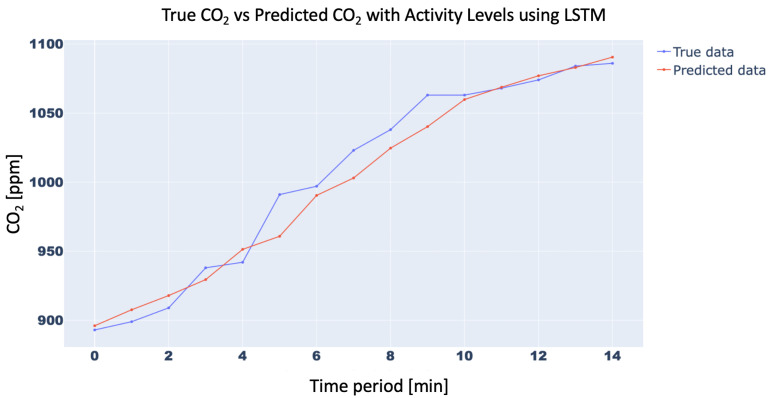
CO${}_{2}$-concentration forecasting results for 15 min with Activity Levels for LSTM.

**Figure 12 sensors-23-05139-f012:**
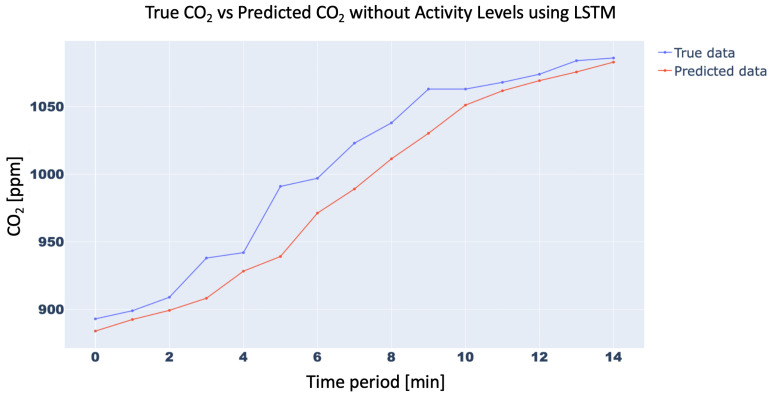
CO${}_{2}$-concentration forecasting results for 15 min without Activity Levels for LSTM.

**Figure 13 sensors-23-05139-f013:**
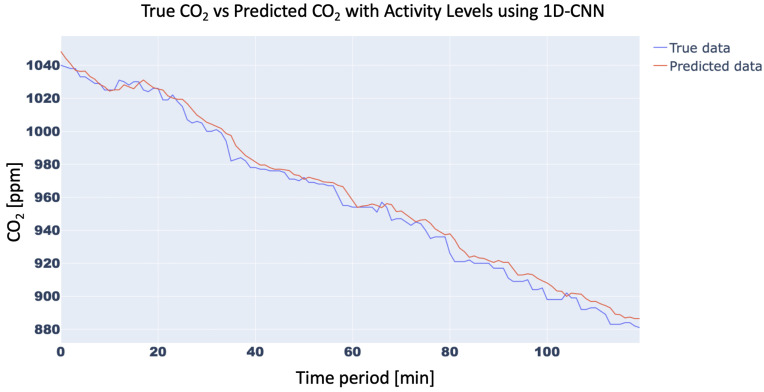
CO${}_{2}$-concentration forecasting results for 120 min with Activity Levels for 1D-CNN.

**Figure 14 sensors-23-05139-f014:**
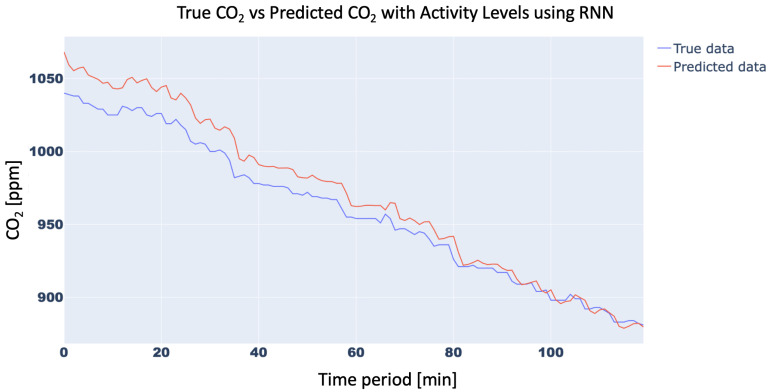
CO${}_{2}$-concentration forecasting results for 120 min with Activity Levels for RNN.

**Figure 15 sensors-23-05139-f015:**
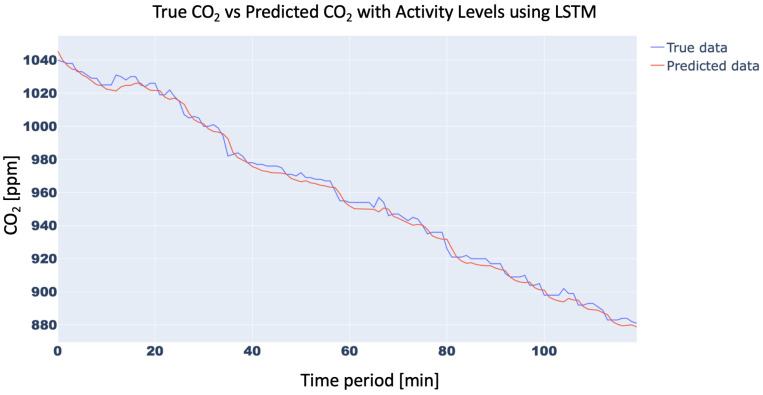
CO${}_{2}$-concentration forecasting results for 120 min with Activity Levels for LSTM.

**Figure 16 sensors-23-05139-f016:**
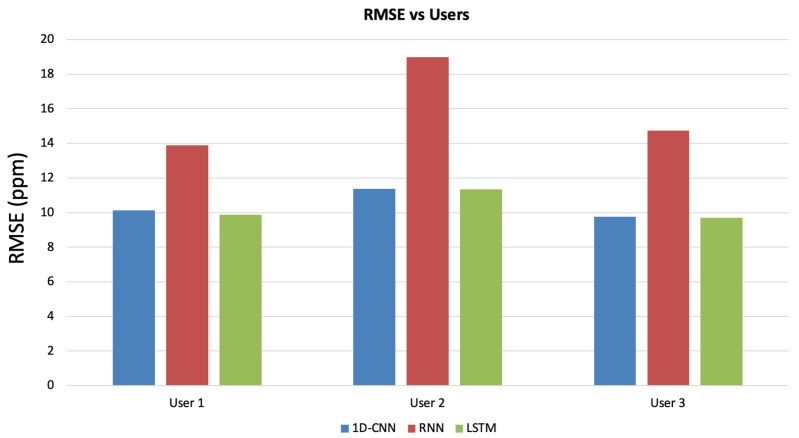
RMSE vs engaged users for the different considered networks.

**Table 1 sensors-23-05139-t001:** Comparison of works on CO${}_{2}$ forecasting in a future window.

Related Work	Learning Dataset Size	Method	Input Parameters	Adaptive	Future Forecasting Window	RMSE
Khazaei et al. [10]	About seven days	Multi Layer Perceptron	CO${}_{2}$, humidity, temperature	No	1 min	17 ppm
Kallio et al. [14]	One year	Ridge regression, Decision Tree, Random Forest, Multi Layer Perceptron	CO${}_{2}$, humidity, temperature, PIR	No	15 min	12–13 ppm
Segala et al. [15]	Thirty days	1D-Convolution Neural Network	CO${}_{2}$, humidity, temperature	Yes	15 min	15 ppm
Presented Work	Ten days	1D-Convolution Neural Network, Recurrent Neural Network, Long Short-Term Memory	CO${}_{2}$, humidity, temperature, wearable accelerometer	Yes	15 min	10–11 ppm

**Table 2 sensors-23-05139-t002:** Range of parameters used for each architecture.

Model	Parameters
1D-CNN	hidden_layer_conv1d = [16, 32, 64, 128, 256],
	hidden_layer_dense = [10, 20, 30, 40, 50, 60],
	number_epochs = [20, 30, 40, 50, 60, 70, 80, 90],
	batch_size = [4, 8, 16, 32, 64, 128],
	dropout = [0.0001, 0.0005, 0.001, 0.005, 0.01, 0.05]
RNN	hidden_layer_simple_rnn = [10, 20, 30, 40, 50, 60],
	hidden_layer_simple_rnn_1 = [10, 20, 30, 40, 50, 60],
	hidden_layer_simple_rnn_2 = [10, 20, 30, 40, 50, 60],
	number_epochs = [20, 30, 40, 50, 60, 70, 80, 90, 100],
	batch_size = [4, 8, 16, 32, 64],
	dropout = [0.1, 0.2, 0.3, 0.4, 0.5]
	dropout_1 = [0.1, 0.2, 0.3, 0.4, 0.5]
	dropout_2 = [0.1, 0.2, 0.3, 0.4, 0.5]
LSTM	hidden_layer_lstm = [10, 20, 30, 40, 50, 60],
	hidden_layer_lstm_1 = [10, 20, 30, 40, 50, 60],
	hidden_layer_lstm_2 = [10, 20, 30, 40, 50, 60],
	number_epochs = [20, 30, 40, 50, 60, 70, 80],
	batch_size = [4, 8, 16, 32, 64],
	dropout = [0.1, 0.2, 0.3, 0.4, 0.5]
	dropout_1 = [0.1, 0.2, 0.3, 0.4, 0.5]
	dropout_2 = [0.1, 0.2, 0.3, 0.4, 0.5]

**Table 3 sensors-23-05139-t003:** Parameters used for each architecture.

Model	Parameters
1D-CNN	optimizer = adam [37], loss_function = mean squared error,
	epochs = 80, batch_size = 128, hidden_layer_conv1d = 128,
	hidden_layer_dense = 20, dropout = 0.005
RNN	optimizer = adam [37], loss_function = mean squared error
	epochs = 80, batch_size = 4, hidden_layer_simple_rnn = 10,
	hidden_layer_simple_rnn_1 = 10, hidden_layer_simple_rnn_2 = 20,
	dropout = 0.1, dropout_1 = 0.3, dropout_2 = 0.3
LSTM	optimizer = adam [37], loss_function = mean squared error,
	epochs = 50, batch_size = 8, hidden_layer_lstm = 60,
	hidden_layer_lstm_1 = 60, hidden_layer_lstm_2 = 30, dropout = 0.1,
	dropout_1 = 0.5, dropout_2 = 0.1

**Table 4 sensors-23-05139-t004:** RMSE and NRMSE at varying degrees of ambiental situation for analyzed architectures.

	Uninhabited Room		Work		Physical Activity	
	RMSE	NRMSE	RMSE	NRMSE	RMSE	NRMSE
1D-CNN	5.89	0.90	6.87	1.39	10.42	0.06
RNN	12.25	1.48	15.56	3.89	15.87	0.09
LSTM	4.85	0.80	5.38	1.34	10.31	0.06

**Table 5 sensors-23-05139-t005:** RMSE for CO${}_{2}$ concentration forecasting for 120 min with and without Activity Levels.

Model	RMSE [ppm]	
	with Activity Level	without Activity Level
1D-CNN	5.17	9.78
RNN	13.23	18.54
LSTM	3.50	7.86

**Table 6 sensors-23-05139-t006:** RMSE for varying inputs for analyzed architectures.

Model	RMSE (ppm)		
	without Temperature	without Umidity	without Temperature and Umidity
1D-CNN	24.08	23.61	28.36
RNN	28.07	27.68	32.73
LSTM	24.43	23.54	27.84

## Data Availability

The data presented in this study are available on request from the corresponding author. The data are not publicly available due to restrictions (their containing information that could compromise the privacy of research participants).

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
