# Peer review of "Multi-Sensor Platform for Predictive Air Quality Monitoring"

_sensors, 2023, doi:10.3390/s23115139_

Round 1
Reviewer 1 Report
The contributions of this manuscript need further refinement, and some of them seem insignificant. please describe the features, functions and innovations of the network in more detail.
In introduction the authors can come up with the existing survey works on the similar topic, probably summary table.
The research method is not clear, please clarify the research method involved.
Why the authors choose the signals that are most widely used in the literature for the analysis of C02 trends over time
To process one-dimensional data, 1D-CNN employs one-dimensional how many convolution layers, pooling layers, dropout layers and activation functions are used must be added.
There are many typos and grammatical errors try to proof read the paper.
LSTM architecture appears without explanation, which makes it difficult for readers to understand.
There should be connection between one section to the next section, the flow is bit harder.
Author Response
We thank the reviewer for the received positive feedbacks.
- Question: In introduction the authors can come up with the existing survey works on the similar topic, probably summary table.
Response: We thank the reviewer for the observation. A table with recently developed works on similar topic was added in the introduction.
- Question: The research method is not clear, please clarify the research method involved.
Response: Thanks for the observation. An adaptive deep learning approach was developed to obtain a system capable of forecasting CO2 concentrations with a good level of accuracy in a future time window of 15 minutes, using a limited dataset for training.
- Question: Why the authors choose the signals that are most widely used in the literature for the analysis of C02 trends over time.
Response: We thank the reviewer for the observation. The best-performing signals were analysed as input for the implemented software framework. The text of the paper was modified to better clarify this aspect.
- Question: To process one-dimensional data, 1D-CNN employs one-dimensional how many convolution layers, pooling layers, dropout layers and activation functions are used must be added.
Response: We thank the reviewer for the observation. In the updated version of the manuscript, the network layers number was described.
- Question: There are many typos and grammatical errors try to proof read the paper.
Response: English was revised and corrected.
- Question: LSTM architecture appears without explanation, which makes it difficult for readers to understand.
Response: We thank the reviewer for the observation. In the updated version of the manuscript, LSTM architecture has been better explained.
Reviewer 2 Report
General Comments
This is a potentially interesting approach, however more details should be provided, and an in-depth analysis of the results performed.
Firstly, no information is presented about validation for the CO2 sensor system or the wearable activity sensor. Based on previous work or manufacturer reported performance, what are the accuracies of these systems? In particular, many low-cost CO2 sensors are known to suffer cross-sensitivities with temperature and humidity; were such cross-sensitivities present in this device, and if so, how were then compensated for?
Second, it would be useful to place the results into a broader context. For instance, the stated goal of the work is to support HVAC operations; is forecasting of CO2 one minute in advance (which appears to be the goal of this paper) appropriate to that task, or would a longer lead-time be required? If HVAC actuation is to be considered, how would that potentially feed back into the forecasting of CO2, and could the proposed architectures handle such a feedback mechanism? Also, while results are presented in units of ppm, it is important to place this into context by reporting the average CO2 concentrations, standard deviation in these concentrations, and the average absolute change in concentrations per minute. You may also consider reporting the performance as a normalized RMSE. Also, comparison of the results from the various DL approaches to a basic forecasting method, e.g., a persistence estimate based on assuming that the CO2 trend observed over the past several minutes will continue, can further allow the additional utility of the DL approach to be shown.
Finally, more insight can be gained by examining the results on a more granular level, e.g., by reporting performance during periods of work versus periods of physical activity versus periods when the room was uninhabited. Looking at performance for each of the three different occupants to determine whether the approach performed better or worse for certain people, and why that might be, could also provide some insight. Performing sensitivity tests with all inputs, not just the physical activity, will help inform which inputs are most needed.
Specific Comments
Title: “Multi-sensory” should be “multi-sensor”.
Line 23: “as the responsible for” should be “as a contributing factor to”.
Line 51: suggest removing “in the”.
Line 75: “Such system” should be “such a system”.
Line 76: “after week” should be “after using a week”.
Line 89: “for home” should be “for a home”.
Line 111: Define “Pc”
Figure 3: Define “Elab. Unit”.
Line 185: “declined” should be “modified” or “customized”.
Line 198: “sequentially” should be “sequence”.
Line s205-206: It is unclear what “the data to be referred is at a longer time point” means; consider rephrasing.
Line 228: Please describe the normalization process (e.g., normalization by the full range, normalization by the standard deviation, or another approach).
Line 254: Please briefly describe the random search procedure, including the dataset holdout method for hyperparameter tuning and the ranges for the parameters considered.
Figure 7: Describe what the annotations and numbers in each box refer to.
Table 1: Please briefly describe or provide a citation for the “adam” optimizer.
Figures 11-19 are very repetitive; consider including only one example, with others moved to a supplemental information if they are needed.
English language is alright, but a final proofreading is recommended.
Author Response
We thank the reviewer for the received positive feedbacks.
- Question: Firstly, no information is presented about validation for the CO2 sensor system or the wearable activity sensor. Based on previous work or manufacturer reported performance, what are the accuracies of these systems? In particular, many low-cost CO2 sensors are known to suffer cross-sensitivities with temperature and humidity; were such cross-sensitivities present in this device, and if so, how were then compensated for?
Response: Thanks for the observation. Information from the manufacturer of the air sensor regarding accuracy and compensation in humidity and temperature has been added to the text, as well as a reference to our previous work on human motion analysis.
- Question: it would be useful to place the results into a broader context. For instance, the stated goal of the work is to support HVAC operations; is forecasting of CO2 one minute in advance (which appears to be the goal of this paper) appropriate to that task, or would a longer lead-time be required? If HVAC actuation is to be considered, how would that potentially feed back into the forecasting of CO2, and could the proposed architectures handle such a feedback mechanism?
Response: The forecast is estimated in a future window of 15 minutes in accordance with other papers in the literature concerning HVAC control. For clarity and simplicity of interpretation, the forecast is plotted minute by minute.
- Question: Also, while results are presented in units of ppm, it is important to place this into context by reporting the average CO2 concentrations, standard deviation in these concentrations, and the average absolute change in concentrations per minute. You may also consider reporting the performance as a normalized RMSE.
Response: We thank the reviewer for the observation. In the updated version of the manuscript, normalized RMSE was added as a metric.
- Question: Finally, more insight can be gained by examining the results on a more granular level, e.g., by reporting performance during periods of work versus periods of physical activity versus periods when the room was uninhabited.
Response: We thank the reviewer for the observation. A table was added showing the performance of the networks considered during periods of work, physical activity and periods when the room was uninhabited.
- Question: Looking at performance for each of the three different occupants to determine whether the approach performed better or worse for certain people, and why that might be, could also provide some insight.
Response: We thank the reviewer for the observation. A graph was added showing the performance of the considered approach during periods of physical activity at varying of the involved users in the trials.
- Question: Performing sensitivity tests with all inputs, not just the physical activity, will help inform which inputs are most needed.
Response: We thank the reviewer for the observation. A series of tests were performed to check the importance of the various inputs and the results are displayed in a Table.
- Question: Title: “Multi-sensory” should be “multi-sensor”.
Response: Done
- Question: Line 23: “as the responsible for” should be “as a contributing factor to”.
Response: Done
- Question: Line 51: suggest removing “in the”.
Response: Done
- Question: Line 75: “Such system” should be “such a system”.
Response: Done
- Question: Line 76: “after week” should be “after using a week”. FATTO
Response: Done
- Question: Line 89: “for home” should be “for a home”.
Response: Done
- Question: Line 111: Define “Pc”.
Response: Done
- Question: Figure 3: Define “Elab. Unit”.
Response: Done
- Question: Line 185: “declined” should be “modified” or “customized”.
Response: Done
- Question: Line 198: “sequentially” should be “sequence”.
Response: Done
- Question: Line s205-206: It is unclear what “the data to be referred is at a longer time point” means; consider rephrasing.
Response: We thank the reviewer for the observation. The sentence has been rephrased.
- Question: Line 228: Please describe the normalization process (e.g., normalization by the full range, normalization by the standard deviation, or another approach).
Response: We thank the reviewer for the observation. The normalization process has been described.
- Question: Line 254: Please briefly describe the random search procedure, including the dataset holdout method for hyperparameter tuning and the ranges for the parameters considered.
Response: We thank the reviewer for the observation. The random search procedure has been described and a table has been added with the ranges for the considered parameters.
- Question: Figure 7: Describe what the annotations and numbers in each box refer to.
Response: Done
- Question: Table 1: Please briefly describe or provide a citation for the “adam” optimizer.
Response: A citation has been added
- Question: Figures 11-19 are very repetitive; consider including only one example, with others moved to a supplemental information if they are needed.
Response: We thank the reviewer for the observation. Not required figures were removed.
Round 2
Reviewer 2 Report
The authors have responded sufficiently to my previous comments. I have a few outstanding comments below:
Line 100: Suggest replacing “recently developed main works” with ”other recent publications” .
Table 1: For consistency, error should be reported either in ppm or percent, not both. A percentage error is more generalizable. Irt should also be specified if this refers to RMSE or some other metric.
Line 113: “as an elaboration unit” can be removed.
Lines 118-120: please provide a citation or reference to the manufacturer’s report with the stated accuracy.
Line 227: Missing space after the sentence.
Line 274: “optimized, defines a search space” should be “optimized, but instead defines a search space”
Table 6: Although mentioned in the caption, NRMSE is not reported in this table.
Figure 16: Suggest replacing “at varying of” with “for the different”.
Minor proofreading and editing is warranted.
Author Response
We thank the reviewer for the received positive feedbacks.
- Question: Line 100: Suggest replacing “recently developed main works” with ”other recent publications” .
Response: Done.
- Question: Table 1: For consistency, error should be reported either in ppm or percent, not both. A percentage error is more generalizable. Irt should also be specified if this refers to RMSE or some other metric.
Response: We thanks the reviewer for the comment. In the revised version of the table, the work reporting the error in percentages was removed, because after a more accurate review, it was not easily comparable with the other works, especially from a methodological point of view. The last column has also been modified indicating the type of error (RMSE).
- Question: Line 113: “as an elaboration unit” can be removed.
Response: Done.
- Question: Lines 118-120: please provide a citation or reference to the manufacturer’s report with the stated accuracy.
Response: Done.
- Question: Line 227: Missing space after the sentence.
Response: Done.
- Question: Line 274: “optimized, defines a search space” should be “optimized, but instead defines a search space”
Response: Done.
- Question: Table 6: Although mentioned in the caption, NRMSE is not reported in this table.
Response: We thank the reviewer for the comment. The misprint in the label of Table 6 has been removed.
- Question: Figure 16: Suggest replacing “at varying of” with “for the different”.
Response: Done.